# A novel cancer immunotherapy using tumor-infiltrating B cells in the APC$^{min/+}$ mouse model

**Xinying Wang, Shohei Asami, Daisuke Kitamura**[ID]*

Division of Cancer Cell Biology, Research Institute for Biomedical Sciences (RIBS), Tokyo University of Science, Noda, Japan

* kitamura@rs.tus.ac.jp

## Abstract

Accumulating evidence has suggested a correlation of tumor infiltrating B cells (TiBcs) and a good prognosis of cancer diseases. In some cases, TiBcs appear to have experienced antigen stimulation since they have undergone class-switching and somatic hypermutation and formed tertiary lymphoid structures around tumors together with T cells. Assuming TiBcs include those that recognize some tumor antigens, we sought to investigate their possible usefulness for cell-mediated immunotherapies. To expand usually small number of TiBcs *in vitro*, we modified our B cell culture system: we transduced B cells with ER$^{T2}$-Bach2 so that they grow unlimitedly provided with tamoxifen, IL-21 and our original feeder cells. Such cells differentiate into plasma cells and produce antibodies upon withdrawal of tamoxifen, and further by addition of a Bach2-inhibitor *in vitro*. As a preliminary experiment, thus expanded splenic B cells expressing a transgenic antigen receptor/antibody against hen egg lysozyme were intravenously injected into mice pre-implanted with B16 melanoma cells expressing membrane-bound HEL in the skin, which resulted in suppression of the growth of B16 tumors and prolonged survival of the recipient mice. To test the usefulness of TiBcs for the immunotherapy, we next used APC$^{min/+}$ mice as a model that spontaneously develop intestinal tumors. We cultured TiBcs separated from the tumors of APC$^{min/+}$ mice as above and confirmed that the antibodies they produce recognize the APC$^{min/+}$ tumor. Repeated injection of such TiBcs into adult APC$^{min/+}$ mice resulted in suppression of intestinal tumor growth and elongation of the survival of the recipient mice. Serum antibody from the TiBc-recipient mice selectively bound to an antigen expressed in the tumor of APC$^{min/+}$ mice. These data suggest a possibility of the novel individualized cancer immunotherapy, in which TiBcs from surgically excised tumor tissues are expanded and infused into the donor patients.

## Introduction

Tumor infiltrating lymphocytes (TIL) are present in the tumor microenvironment. Numerous reports have shown association between the number of TIL and the patient's chance for

**Data Availability Statement:** All relevant data are within the manuscript and its Supporting information files.

**Funding:** This study was supported by Grant-in-Aid for Scientific Research on Innovative Area,

17H05803 and 19H04818 (DK), from the Ministry of Education, Culture, Sports, Science and Technology in Japan (https://www.mext.go.jp/en/); and Takeda Science Foundation (https://www.takeda-sci.or.jp). The funders had no role in study design, data collection and analysis, decision to publish, or preparation of the manuscript.

**Competing interests:** The authors have declared that no competing interests exist.

survival in various types of cancers [1]. But TILs are induced to become dysfunctional in the tumor microenvironment, resulting in the abrogation of antitumor immune responses [2]. Although much of tumor immunology has been focused on CD8$^+$ cytotoxic T cells whose activity has been shown to be closely related to patient survival [3], other types of cells must reasonably collaborate with CD8$^+$ T cells. It was reported that B cells account for 25% of all lymphocytes in some tumors [4], and as much as 40% in some breast cancer subjects [2, 5]. B cells infiltrating in tumor tissues (tumor-infiltrating B cells, TiBcs) are thought to be involved in anti-tumor immunity. Correlation between the number of TiBcs and survival of the disease patients has been reported for various cancers, including cutaneous melanoma, breast and ovarian cancer [6]. Since TiBcs often express class-switched and mutated antigen receptors, they are considered as antigen-activated and affinity-maturated memory B cells (MBCs) that may have recognized cell surface or intracellular tumor antigens [7]. Accordingly, antibodies (Abs) against tumor antigens have been frequently found in the serum of cancer patients [8]. For example, studies have shown that approximately 50% of breast cancer patients develop antibody (Ab) responses to primary tumors [5] and that levels of antigen/Ab complex in the serum are higher in breast cancer patients compared to healthy controls [5, 9].

It is assumed that immune responses to autologous tumors are initially driven by innate signals in the tumor microenvironment, which supposedly promotes B cell infiltration into the tumor tissues [10]. It is assumed that, if such B cells recognize antigens expressed by the tumor cells, they proliferate, undergo immunoglobulin gene mutations, and develop into MBCs, perhaps with the aid of helper T cells that have also been developed in response to the tumor antigens. The clonally expanded MBCs may reside in the tumor tissue and occasionally respond to their cognate tumor antigens to produce tumor-specific Abs. Such MBCs may also work as antigen presenting cells and promote T cell response to the tumor [10]. As some of TiBcs and Abs thereof may specifically bind and respond to cancer cells of the donor patient and therefore should be applicable to individualized immunotherapy.

Keeping such applications of TiBcs in mind, we previously developed a culture system to efficiently expand B cell *in vitro*, namely, *in-vitro*-induced germinal center B (iGB) cell culture system. In this system, B cells from tissues or blood are cultured on a feeder layer of 40LB cells, a mouse fibroblast line expressing exogenous CD40L and BAFF [11]. In the original protocol, mouse splenic naïve B cells are successively cultured with IL-4 and IL-21, each for 4 days, which resulted in some 10,000-fold increase of the number of B cells. The proliferating B cells (termed iGB cells) phenotypically and functionally resemble genuine GC B cells, class-switched to either IgG1 or IgE, expressing GC-signature transcription factors such as Bcl-6 and Bach2, although they do not undergo somatic mutation of immunoglobulin genes. During the culture with IL-21, the iGB cells gradually differentiate into plasma cells, due to gradual expression of Blimp1, a master regulator of B cell differentiation into plasma cells. The growth of iGB cells usually stops by day 10 of the culture probably due to their overall differentiation. However, we have established that exogenous Bach2 expression allows their limitless growth, provided with the replenishment of feeder cells and IL-21, based on the report that Bach2 suppresses the expression of gene encoding Blimp1 [12]. It has also been found that pre-activated B cells such as GC and memory B cells can be cultured without initial supplement of IL-4 but starting with IL-21 (our unpublished data).

When iGB cells are cultured with IL-4 and IL-21 successively, and those deprived of IgE$^+$ and CD138$^+$ cells are transferred into mice, such cells differentiate into plasma cells *in vivo* and produce Abs, mostly of IgG1, for as long as a month [11, 13]. Thus, we sought to utilize this iGB cell transfer as a new cancer immunotherapy, which would supply Ab-producing cells to patients instead of Abs. To test this possibility in mice as a model, we generated mouse

melanoma cell line B16 expressing membrane-anchored from of hen egg lysozyme (mHEL) [14] as a surrogate tumor antigen, and generated iGB cells from spleen B cells of Hy10 mice, which carry a hen egg lysozyme (HEL)-specific heavy chain (VDJ9) and light chain (κ5) genes in knock-in and transgenic configurations, respectively [15]. The mHEL-expressing B16 (B16-mHEL) cells colonized and formed numerous numbers of tumors in lungs three weeks after intravenous transfer, as the original B16 cells would do. When the Hy10-derived iGB (Hy10-iGB) cells were injected intravenously (i.v.) on the next day of the B16-mHEL injection, anti-HEL IgG Abs were produced *in vivo* and the tumor formation in lungs was almost completely suppressed, whereas iGB cells derived from normal C57BL/6 (B6) mice hardly suppressed it [13].

In the present study, we further developed the iGB-cell-mediated immunotherapy model. We first tested whether the Hy10-iGB cell injection inhibit progression of tumors of B16-mHEL cells pre-implanted subcutaneously (s.c.) in mice. We then used APC$^{min/+}$ mice, a model of human genetic disease familial adenomatous polyposis (FAP), as a model that spontaneously develops tumors [16, 17]. We collected TiBcs from intestinal polyps of APC$^{min/+}$ mice, cultured them as iGB cells, and then injected into adult APC$^{min/+}$ mice regularly to examine whether the TiBc-derived iGB cells can ameliorate tumor progression in these mice.

## Materials and methods

### Mice

C57BL/6 (B6) mice were purchased from Japan SLC. APC$^{min/+}$ mice (a gift of Dr. Iwakura) had been backcrossed to the congenic C57BL/6 strains. Mice 8–10 weeks of age were used for experiments unless indicated otherwise. All mice were maintained in our mouse facility under specific pathogen-free conditions. When we dissected the mice, mice were killed by cervical dislocation under anesthesia with Isoflurane in all mouse experiments. In survival studies, mice were monitored daily for general appearance and weighed every week. When mice show signs of losing ability to ambulate, they were euthanized as above. All animals were treated under the protocols approved by the Animal Care and Use Committee of the Tokyo University of Science (ACUC-TUS), by researchers who were trained for animal care through a lecture held by ACUC-TUS.

### Cell line

B16-mHEL-GFP mouse melanoma cells [13] and 40LB [11] were maintained in D-MEM medium (high glucose; Wako) supplemented with 10% FBS, 100 units/ml penicillin, and 100 μg/ml streptomycin (GIBCO) in a humidified atmosphere at 37˚C with 5% $CO_2$.

### Isolation and culture of primary B cells

Naïve B cells were purified from the spleens of Hy10 mice by 2-step negative sorting: cells were stained with biotinylated monoclonal Abs (MoAbs) against CD43 (BD Pharmingen, S7), CD4 (BioLegend, GK1.5), CD8α (BioLegend, 53–6.7), CD49b (BioLegend, DX5), Ter-119 (BioLegend, TER-119), and then streptavidin-particle-DM (BD Biosciences), passed first through an iMag column (BD Biosciences) and unbound cells were then passed through a MACS LS column (Miltenyi Biotec). The unbound cells were mostly B cells (>97% purity). Purified naïve B cells were cultured in B cell medium (BCM) on a feeder layer of irradiated 40 LB cells with IL-4 and IL-21, sequentially, to generate iGB cells [11].

TiBcs were purified from the tumor of the APC$^{min/+}$ mice. Intestinal tumors were cut off and incubated in a small bottle with 10 ml of pre-warmed collagenase buffer for 20 min at

37˚C with continuous agitation. The dispersed cells were put into a 15 ml tube, into which 4 ml of 40% Percoll buffer was added and mixed by pipetting and vortex. Then 4 ml of 80% Percoll buffer was gently added under the 40% Percoll buffer. The tube was then centrifuged (2000 rpm for 20 min), and a middle layer about 2–3 ml was collected into a new 15 ml tube. The tube was filled with 10 ml of 5% FCS/RPMI and centrifuged (1400 rpm for 7 min at 4˚C). Thus isolated TiBcs were cultured in BCM on a feeder layer of irradiated 40 LB cells with IL-21 to generate iGB cells.

B cells were purified from normal intestinal tissues of mice as follows: small intestines were cut into pieces and put into a small bottle, into which 10 ml pre-warmed EDTA buffer was added. The bottle was incubated for 20 min at 37˚C with continuous agitation. The following steps were the same as the method to purify and culture TiBcs as described above.

To remove 40LB cells from iGB cells after the culture, the cells from the culture were stained with biotinylated anti-H-2Kd (BioLegend, SF1-1.1) Ab and Streptavidin Particle Plus DM, followed by negative sorting using the iMag system (BD Biosciences).

## Retroviral transduction

For production of retrovirus, pMXs-$ER^{T2}$-Bach2-ires-CD8α vector, encoding a mutated estrogen receptor hormone-binding domain ($ER^{T2}$) fused with human Bach2 ($ER^{T2}$-Bach2) and mouse CD8α as an infection marker, were transfected into Plat-E cells by PEI (Mw 40,000; Polysciences). The virus-containing supernatant was harvested 2 days after transfection and added to day 3 iGB cells. Cells were spin-infected at 2000 rpm, 37˚C for 90 min with 10 μg/ml DOTAP Liposomal Transfection Reagent (Roche). One day later, cells were harvested, seeded on new 40LB feeder layers, and cultured in BCM with IL-21.

## Flow cytometry

TiBcs were stained with various combinations of the following Abs: FITC-, PE-, PE-Cy7-, PerCy5.5- and APC Cy7-conjugated anti-mouse IgG (Southern Biotech, 1030–02), IgA (Southern Biotech, 1165–11), IgM (eBioscience, 11/41), IgD (BioLegend, 11-26c.2a) and B220 (BioLegend, RA3-6B2). Cells were stained with propidium iodide, just before analysis, to eliminate dead cells in the data analyses. When the iGB cells were analyzed, 40LB feeder cells were gated out based on FSC versus SSC. Analysis was performed by FACS Calibur or Canto II (BD Bioscience). The results were analyzed by FlowJo (Tree Star, Inc.).

## Antibody production from iGB cells in culture

$ER^{T2}$-Bach2-transduced iGB cells were transferred onto a feeder layer of irradiated 40LA cells (40LB cells stably transfected with mouse APRIL expression vector) and further cultured in BCM supplemented with heme (10 μM), IL-5 (10 ng/ml) and IL-6 (10 ng/ml) for 5 days.

## ELISA

Flat-bottom 96-well plate (Nunc) was coated with 1 mg/ml of anti-mouse IgA (Southern Biotech, 1040–01), IgM (Southern Biotech, 1020–01) or IgG (Southern Biotech, 1030–01) by incubating at room temperature for 1 hour, and then blocked with 3% BSA or 5% FCS. The diluted supernatants of B cell culture were added to the wells and allowed to react for 1 hour at room temperature. As a standard, Mouse Reference Serum (Bethyl, RS10-101) was 2-fold serially diluted. Abs bound to the plate were detected with 1 mg/ml of HRP-conjugated goat anti-mouse IgA (Southern Biotech, 1040–05), IgM (Southern Biotech, 1020–05) or IgG (Southern Biotech, 1033–05) after 1 h incubation. Then Abs bound to the plate were developed with

TMB substrate (Sigma) and the reaction was terminated by sulfuric acid. Absorbance values were measured at 450 nm using plate reader (Bio-Rad).

## Immunofluorescence microscopy

Freshly isolated normal intestinal tissues and tumor tissues were snap-frozen in OCT compound (Sakura Finetek) and 5 to 6-μm-thick sections were prepared using a cryostat. After drying at room temperature, sections were fixed in acetone for 10 min and then allowed to dry. Nonspecific binding was blocked with 3% BSA/PBS for 30 min at room temperature, and then the slides were washed and stained for 1 h at room temperature with mouse anti-E-Cadherin (BD Transduction Laboratories, 610182) or supernatants of iGB cell culture or mouse sera, appropriately diluted as indicated. After washing, the slides were stained for 1 h at room temperature with Alexa Fluor® 660 goat anti-mouse IgG(H+L) (Invitrogen, A21054) and anti-mouse B220-FITC (BioLegend, RA3-6B2) or anti-mouse IgG-FITC (Southern Biotech, 1030–02), respectively. After washing, slides were mounted with the fluorescent mounting medium (ProLong™ Gold antifade reagent with DAPI, Invitrogen) and examined using an immunofluorescence microscope (BZ-9000; Keyence).

The fixed sections were also stained with haematoxylin and eosin (H&E) through gradual changes of alcohols for dehydration.

## Immunohistochemistry

Freshly isolated normal intestinal and tumor tissues were snap-frozen in OCT compound (Sakura Finetek) and 5 to 6-μm-thick sections were prepared using a cryostat. After drying at room temperature, sections were fixed in acetone for 10 min and then allowed to dry. Then sections were treated in 95˚C sodium citrate buffer for 15 min to restore antigens. Nonspecific binding was blocked with 3% BSA/PBS for 30 min at room temperature, and then the slides were washed and stained for 1 h at room temperature with antibodies against cleaved caspase 3 (Cell Signaling, D175) or Ki67 (Cell Signaling, D3B5), appropriately diluted as indicated. After washing, the slides were stained for 1 h at room temperature with anti-rabbit-HRP. After washing, slides were mounted with the DAB (Dako) and counter stain with hematoxylin, examined using an immunofluorescence microscope (BZ-9000, Keyence).

## Western blotting

Total proteins were extracted from normal intestinal tissues and tumor tissues. Equivalent amounts of proteins were subjected to SDS-PAGE and transferred to a PVDF membrane. The blots were probed with the indicated sera or anti-GAPDH (MBL, 3H12) as the primary Ab, followed by incubation with HRP-conjugated goat anti-mouse IgG (Southern Biotech,1033–05) as the secondary Ab. Immunoreactive protein bands were visualized using the enhanced chemiluminescence detection kit (Sigma-Aldrich) on a ChemiDoc MP imaging system (Bio-Rad).

## Statistical analysis

Statistical analysis was performed using the Student's *t* test as appropriate. $p < 0.05$ was considered statistically significant. To assess survival rate, the Kaplan-Meier model was used and comparison of survival between groups was performed using the LogRank test with GraphPad Prism software (GraphPad Software, San Diego, CA).

## Results

### Antigen-specific iGB cells suppress growth of pre-implanted melanoma tumor

Previously, we demonstrated that infusion of HEL-specific iGB cells (Hy10-iGB) inhibited metastatic growth of B16 melanoma cells expressing membrane-bound HEL (B16-mHEL) that were transferred i.v. into mice at the same timing and prolonged survival of the mice [13]. To further verify the efficacy of the iGB-cell-mediated cancer therapy in a mouse model, we examined whether infusion of iGB cells suppress the growth of pre-existing tumor. For this experiment, we retrovirally transduced Hy10-iGB cells with ER$^{T2}$-Bach2, which enable the cells to grow unlimitedly on 40LB feeder cells provided with IL-21 and tamoxifen. Bach2 is known to suppress the expression of Blimp1, a master regulator of plasma cell differentiation, and tamoxifen-induced Bach2 activation allows unlimited expansion of iGB cells (to be reported elsewhere).

We injected the Hy10-iGB cells (or PBS alone as a control) i.v. into B6 mice that had been implanted s.c. with B16-mHEL cells seven days previously (Fig 1A). B16-mHEL cells formed a skin tumor of readily visible size (~5 mm in diameter) by the time of the Hy10-iGB cell injection. In the mice injected with Hy10 iGB cells, serum IgG1 concentration was apparently increased by day 7 (Fig 1B) as expected with our previous report [13], considering that the cultured Hy10 iGB cells were mostly IgG1$^+$ and must have differentiated into plasma cells *in vivo* being free of tamoxifen. At 10 days after injection of Hy10-iGB cells, the size of the B16-mHEL cell tumor was significantly smaller as compared to mice injected with PBS alone (Fig 1C).

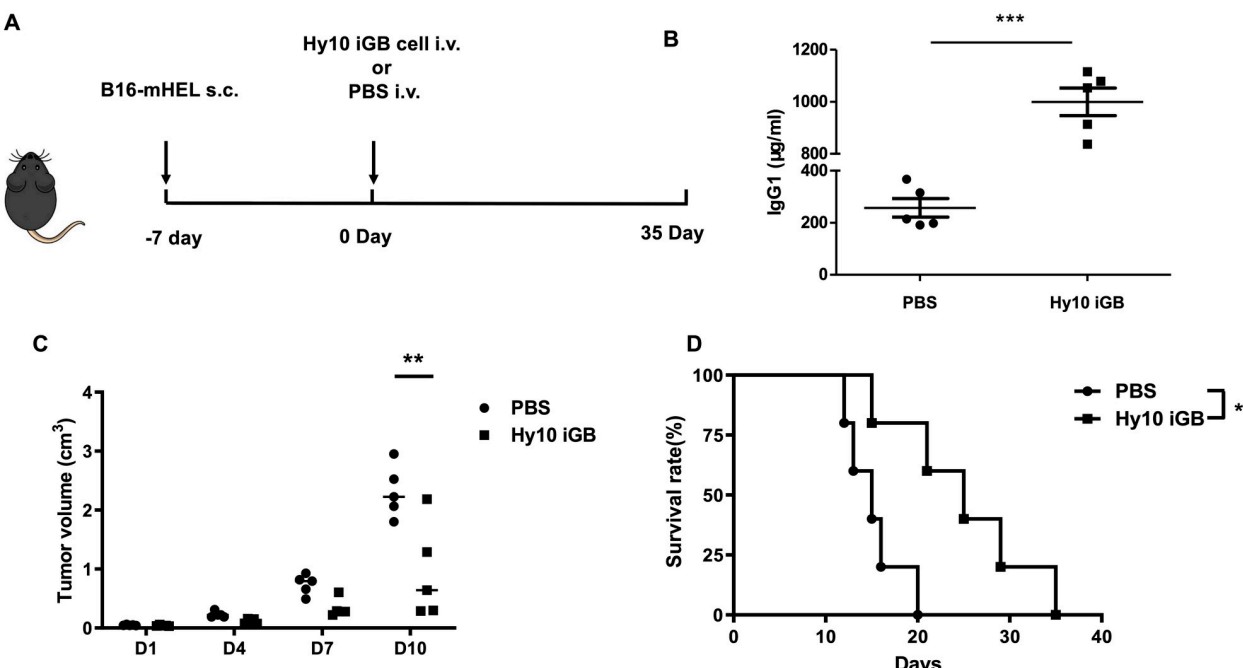

**Fig 1. Hy10-iGB cells suppressed pre-existing B16-mHEL cell growth *in situ*.** (A) The experimental strategy. B16-mHEL-GFP cells (5×10$^5$ cells/ mouse) were transferred s.c. into C57BL/6 mice on day -7. On day 0, ER$^{T2}$-Bach2-transduced iGB cells (1×10$^7$ cells/mouse) derived from splenic B cells of Hy10 mice (Hy10-iGB) or PBS alone were transferred i.v. into the mice. (B) Concentration of IgG1 in the sera of the mice on day 7 was determined by ELISA. n = 5. ***p<0.001. (C) The size of tumors in the mice in (B) was monitored until day 10. The volume of the tumors was calculated using the equation: V = 4π(L1xL2$^2$)/3 cm$^3$, where V = volume (cm$^3$), L1 = the longest radius (cm), L2 = the shortest radius (cm). n = 5. **p<0.01. (D) Survival rate of mice was compared using LogRank test. n = 5. *p<0.05. All data are representative of two independent experiments.

Long-term observation of these mice revealed that the mice injected with Hy10 iGB cells survived significantly longer than those with PBS alone (Fig 1D). These data indicated that HEL-specific Abs produced by iGB-cell-derived plasma cells *in vivo* inhibited growth and perhaps metastasis of the pre-implanted B16-mHEL cells in the recipient mice, and imply a possible clinical application of human iGB cells for a cancer therapy.

## Production of tumor-specific antibody from TiBc-derived iGB cells

To apply iGB cells for a cancer therapy, B cells recognizing some tumor antigens would be needed. Such B cells are likely very few in peripheral blood of a cancer patient but might be enriched in TiBcs. Expecting to utilize TiBcs from surgically resected specimens for the future application, we used APC$^{min/+}$ mice as a model system. APC$^{min/+}$ mice carry a *Min* (multiple intestinal neoplasia) mutant allele of the *APC* (adenomatous polyposis coli) locus that encodes a nonsense mutation at codon 850. Like humans with germline mutations in APC, APC$^{min/+}$ mice are predisposed to intestinal adenoma formation [16], thus providing a good animal model for studying the role of intestinal tumorigenesis [17].

We first histologically analyzed the intestine of APC$^{min/+}$ mice for the presence of TiBcs. We found small clusters of B cells sparsely scattered mainly in the interstitial regions of the tumor and in the lamina propria of normal mucosa (Fig 2A). To utilize the TiBcs in the tumor of APC$^{min/+}$ mice, we isolated B cells from intestinal tumors of APC$^{min/+}$ mice and those from normal intestinal tissues of APC$^{+/+}$ mice for a control. We cultured these B cells on the 40LB feeder layer with IL-21 (iGB cell culture) and transduced them with ER$^{T2}$-Bach2. In a typical case, the proportion of IgG$^+$ cells was low in the TiBcs before the culture (day 0), but the number of them increased for about 14-fold of the starting number by day 7, whereas the number of IgM$^+$ and IgA$^+$ cells increased for about 2- and 5.8-fold, respectively (Fig 2B and 2C). It remains elusive whether this result is due to a selective growth of IgG$^+$ cells or to *de novo* class switching to IgG in the iGB cell culture.

To obtain Abs from expanded iGB cells, we withdrew tamoxifen from the culture and further added heme to inactivate Bach2 function to allow these iGB cells to differentiate into plasma cells producing Ab (the system will be reported elsewhere). Heme was previously reported to bind to Bach2 to inhibit its DNA-binding activity, leading to Blimp1 induction in B cells [12]. iGB cells of normal intestinal tissue B cells from APC$^{+/+}$ mice (Int. B-iGB) and those from TiBcs from APC$^{min/+}$ mice (TiBc-iGB) were cultured on feeder layers of 40LA cells (40LB cells expressing APRIL) with heme, IL-5 and IL-6 for 5 days. IgG production was efficient in this culture system, whereas IgA and IgM production was minimum (Fig 2D). The supernatants on day 5 were used as primary Abs for immunofluorescent staining of the sections of intestines including tumors of APC$^{min/+}$ mice. The results showed that the IgG Ab from TiBc-iGB cells could stain a tumor area, but not a normal area, of the intestinal tissue, while IgG Ab from Int. B-iGB cells did not significantly stained either areas (Fig 2E). These data indicate that TiBcs from APC$^{min/+}$ mice tumor included B cells that are specific to some antigens expressed in the tumor and thus likely to have responded to such antigens *in situ*.

## TiBc-derived iGB cells inhibit intestinal tumor growth in APC$^{min/+}$ mice *in vivo*

The results of these *in vitro* studies suggested that Abs produced from TiBc-iGB cells may suppress tumor growth in APC$^{min/+}$ mice. The propagated TiBc-iGB or Int. B-iGB cells, prepared as above, or PBS alone, were transferred i.v. into 8-week-old APC$^{min/+}$ mice, at the timing when the intestinal tumors become visible in most APC$^{min/+}$ mice, and the injection was repeated every two weeks for 8 weeks (Fig 3A).

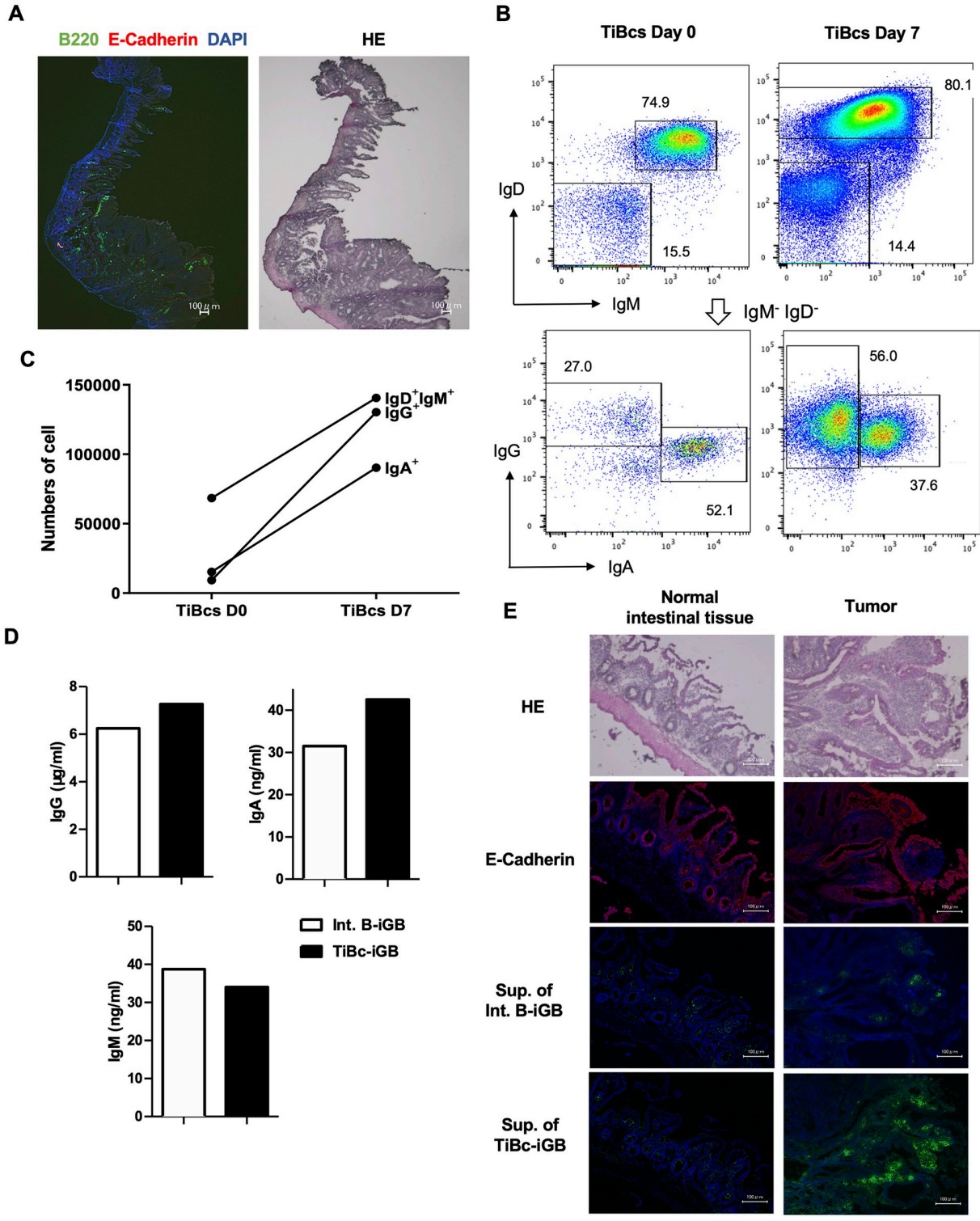

**Fig 2. Antibody produced by APC^{min/+} TiBcs binds to original tumor tissue.** (A) Representative data of immunofluorescent staining. Sections of intestinal tissue of a 20-week-old APC^{min/+} mouse were stained with hematoxylin/eosin (HE, right), or with Abs against B220 (green) and E-Cadherin (red) together with DAPI (blue, left). (B) TiBcs of an APC^{min/+} mouse were transduced with ER^{T2}-Bach2 and cultured on γ-irradiated 40LB cells in BCM with IL-21 and tamoxifen. Before (day 0) and on day 7 of the iGB cell culture, B cells were analyzed for expression of Ig classes by flow cytometry. Th data represents the cells within the lymphocyte gate defined by side- and forward-scatter and gated as B220^+. A number at each window indicates the percentages of the B cells (top) or of the gated IgM⁻IgD⁻ cells (bottom). (C) The

numbers of IgG[+], IgA[+] and IgD[+] IgM[+] cells on day 0 and day 7 of the culture, estimated by the data in (B). (D) Day 7 iGB cells derived from mixed normal intestinal tissue B cells from three APC[+/+] mice (Int. B-iGB) and those from mixed TiBcs from three APCmin/+ mice (TiBc-iGB) were transferred onto 40LA feeder cells and further cultured in the medium without tamoxifen with addition of heme, IL-5 and IL-6, for 5 days. Concentration of IgG, IgA and IgM in the supernatants was estimated by ELISA. (E) A representative data of immunofluorescent staining. Serial sections of intestinal tissue of a 20-week-old APC[min/+] mouse were stained with hematoxylin/eosin (HE), with anti-E-Cadherin Ab (red), or with culture supernatants (diluted to contain 1 µg/ml IgG) of the Int. B-iGB cells or the TiBcs-iGB cells as the primary Ab, followed by anti-mouse IgG-FITC (green) as the secondary Ab, as indicated. For the immunofluorescent analysis, sections were simultaneously stained with DAPI (blue). Images (merged with DAPI image for immunofluorescent staining) of normal areas (left) and tumor areas (right) are shown. All data are representative of two independent experiments.

Long-term observation of these mice revealed that the APC[min/+] mice injected with PBS alone or the Int. B-iGB cells started to die in one or three weeks after the last injection, respectively, and all of them died within 20 weeks in the two groups. By contrast, mice transferred with TiBc-iGB cells survived much longer and 43% of them survived for longer than 30 weeks (Fig 3B). Accordingly, bodyweight loss was mitigated in the mice transferred with TiBc-iGB cells as compared to those with Int. B-iGB cells or PBS alone (Fig 3C). Intestines of mice in a different set of the same experiment were inspected at 16 weeks of age. The intestines of the mice that received Int. B-iGB cells or PBS alone had a numerous number of tumors. By contrast, there were markedly fewer number of tumors, especially of larger sizes (longer than 3

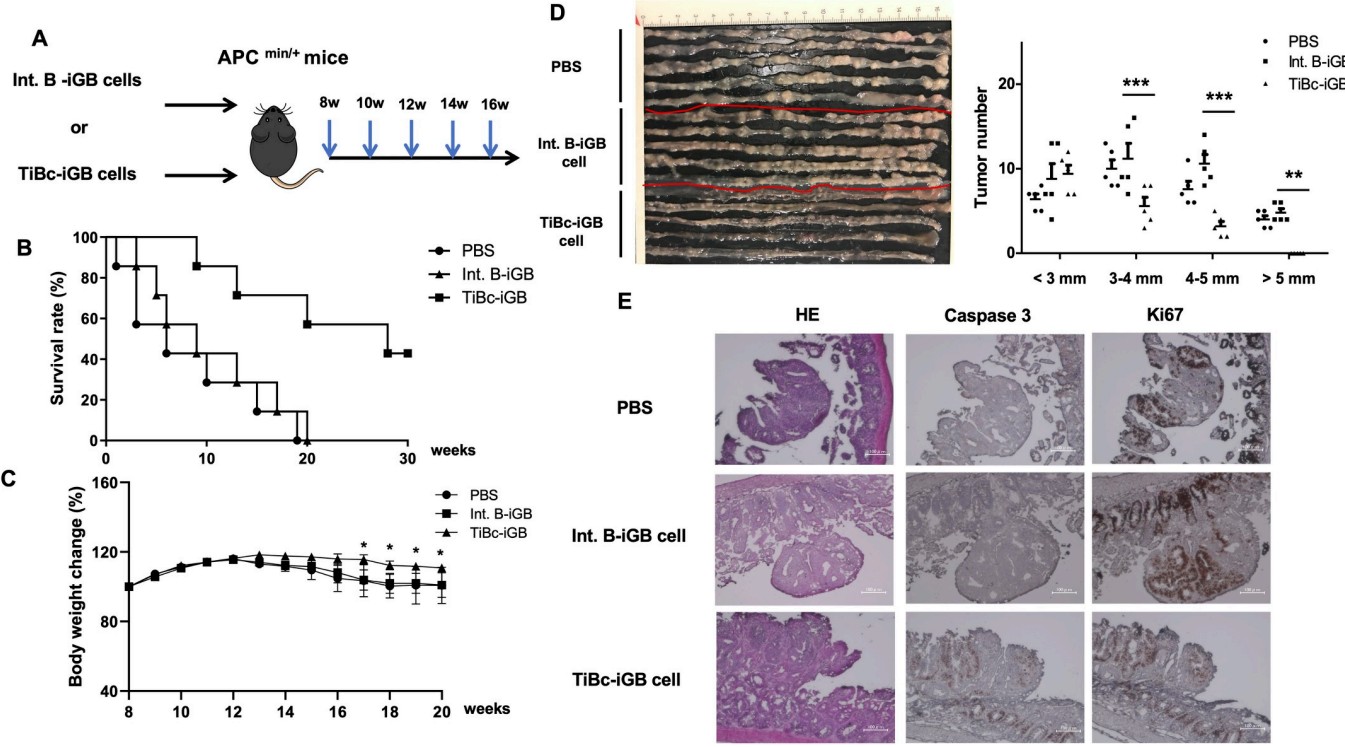

**Fig 3. APC[min/+]-derived TiBcs inhibit tumor progression in APC[min/+] mice *in vivo*.** (A) Experimental strategy. The Int. B-iGB cells or TiBc-iGB cells used in Fig 2D (5×10[6] cells/mouse) or PBS alone were injected i.v. into 8-week-old APC[min/+] mice five times at every two weeks. (B) Survival rates of the mice in the three groups treated as in (A) after the last injection of iGB cells (week 0), were compared using LogRank test, n = 7. *p<0.01. (C) Body weights of survived mice in (B) were monitored until 20 weeks of age and changes of body weights from those in 8 weeks of age (divided by the weight at 8-week) were expressed as percentages. Data are shown as mean ±SEM *P < 0.05, **P < 0.01. (D) Photographs of the intestines of the APC[min/+] mice treated as in (A) taken at 16-weeks of age (left), n = 5. The number of tumors within the indicated range of diameter (mm) per each intestine, counted on the photographs, is plotted (right). Data are expressed as mean +SEM **P<0.01, ***P<0.001. (E) Representative data of immunohistochemistry staining. Serial sections of intestinal tumor areas of intestines from the APC[min/+] mice, each one selected from the three groups in (D), were stained with hematoxylin/eosin (HE), or with Abs against cleaved caspase 3 or Ki67, as indicated. All data are representative of two independent experiments.

mm in diameter), in the intestines of mice that had received TiBc-iGB cells (Fig 3D). Immuno-histochemistry staining of the tumor tissues revealed that somewhat increased expression of active (cleaved) caspase3 and decreased expression of Ki67 in the tumors of the mice transferred with TiBc-iGB cells, as compared with those with Int. B-iGB cells or PBS alone (Fig 3E). Collectively, these data strongly suggest that tumor-specific Abs produced in-vivo by TiBc-iGB cell-derived plasma cells inhibited tumor growth in the intestine and prolonged survival of the recipient APC$^{min/+}$ mice.

## Sera of TiBc-iGB cell-transferred mice bind to antigens of APC$^{min/+}$ mouse tumors

In the setting of the experiment shown in Fig 3, Abs produced in vivo from the transferred iGB cells and those produced by the host cannot be distinguished owing to their syngeneic origins, except by showing the tumor-specificity of the TiBc-iGB-cell-derived Abs, if any. Thus, sera of APC$^{min/+}$ mice transferred with TiBc-iGB cells, Int. B-iGB cells or PBS alone were examined for their binding to intestinal tissues of APC$^{min/+}$ mice. The immunofluorescence microscopy demonstrated that the serum IgG from the mice transferred with TiBc-iGB cells bound to a tumor area but scarcely to a normal area of the same intestinal tissue, whereas serum IgG from those transferred with Int. B-iGB cells or PBS alone little bound to the tumor and the normal areas (Fig 4A). Thus, TiBc-iGB cells appear into differentiate to plasma cells in vivo and to produce Abs which selectively recognize antigens on tumor tissues.

Above result was further evaluated by Western blot analysis against proteins extracted from intestinal tumors of APC$^{min/+}$ mice or those from normal intestine of APC$^{+/+}$ mice. IgG Abs in the sera from APC$^{min/+}$ mice transferred with TiBc-iGB cells, Int. B-iGB cells or PBS alone commonly detected several bands in the extract of the tumor, but the those with TiBc-iGB cells selectively detected a band of about 20 kDa, which was not detected in the extract of the normal intestine (Fig 4B). Collectively, these data indicate that Abs produced *in vivo* from the transferred TiBc-iGB cells can recognize a certain antigen that is specifically present in the tumor tissue of APC$^{min/+}$ mice.

## Discussion

There is increasing evidence that TiBcs play complex functions in tumor immunity. On the one hand, TiBcs have the function of promoting tumors by producing IL-10, TGF-β, and IL-35; by the lymphotoxin/ IKKa-BMI1 signaling pathway; or by directly promoting tumor progression through engaging PD-L1/PD-1 molecules [18, 19]. On the other hand, TiBcs can inhibit tumor development through Ab production, serving as antigen presenting cells, and secreting antitumor cytokines [20]. Recent clinical research demonstrated that in patients with immunotherapy, tumor infiltrating B cells are positively associated with improved survival [21–24]. Moreover, TiBcs were reported to decrease tumor cells viability by activating T cells and NK cells. It has been shown in a mouse model that depletion of B cells enhances tumor growth in hepatocellular carcinoma [25]. These contradictory observations may be partly due to the heterogeneity and different subpopulations of B cells in different tumor microenvironments (TME) under various therapies [26]. It was found that a special subset of B cells defined as regulatory B cells increased in TME and was associated with progression of various cancers [25]. Despite mixed views on the role on TiBcs, it is assumed that most of the TiBcs recognized some tumor-associated antigens before developing into various subsets of tumor-residing B cells, and therefore their B-cell receptor/Ab repertoire may be specific to tumors and thus useful for Ab-mediated immunotherapy.

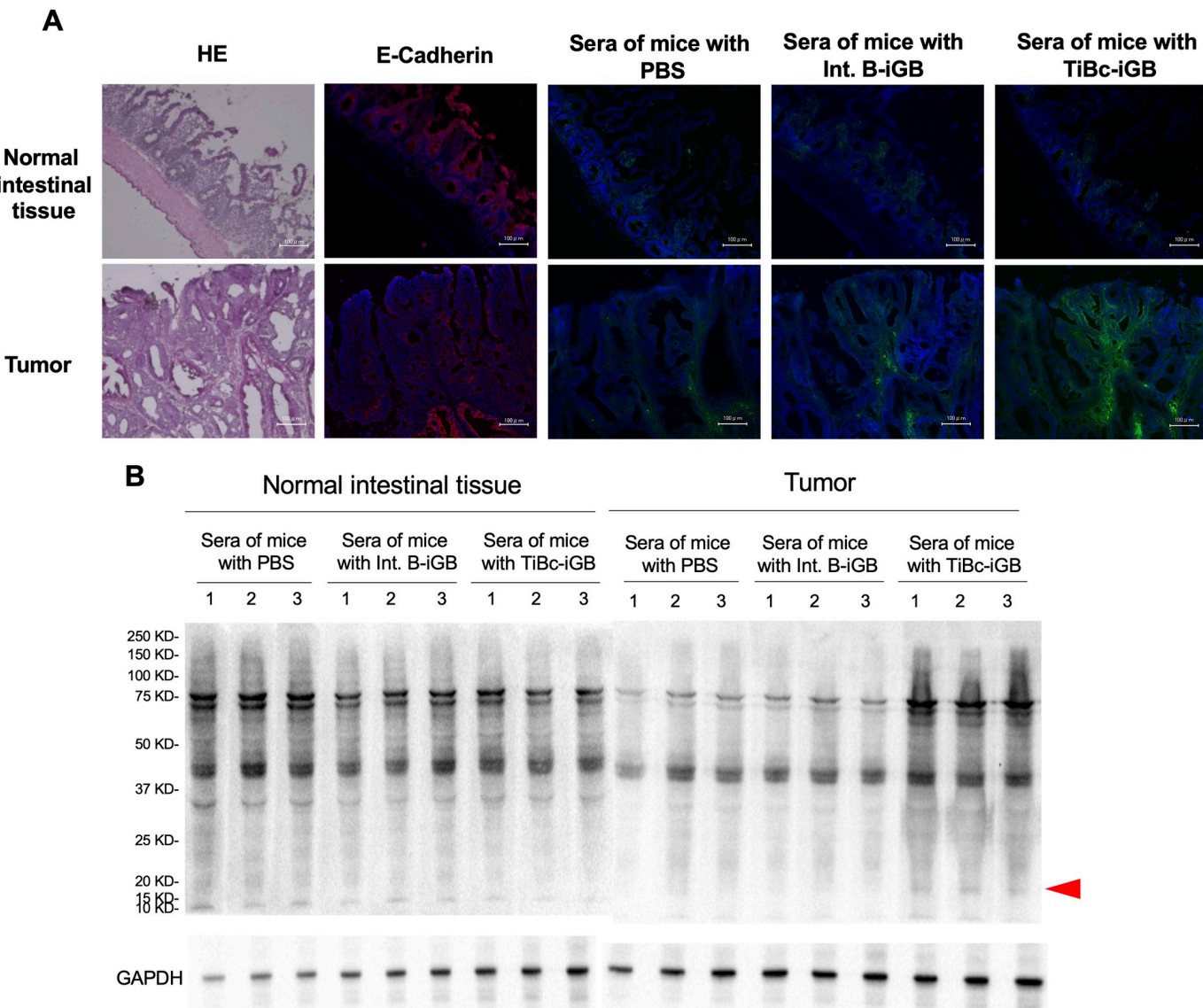

**Fig 4. Serum antibodies of TiBc-injected mice recognize APC$^{min/+}$ tumor antigens.** (A) Representative data of immunofluorescent staining. Serial sections of intestinal tissue of a 20-week-old APC$^{min/+}$ mouse were stained with hematoxylin/eosin (HE), with anti-E-Cadherin Ab (red), or with undiluted sera of mice having been transferred with PBS alone, Int. B-iGB cells or TiBc-iGB cells (mixture of three randomly selected sera from mice in each group in Fig 3B) as the primary Abs, followed by anti-mouse IgG-FITC (green) as the secondary Ab, as indicated. For the immunofluorescent analysis, sections were simultaneously stained with DAPI (blue). Images (merged with DAPI image for immunofluorescent staining) of normal area (top row) and tumor area (bottom row) are shown. (B) Combined extracts from normal intestinal tissues of two APC$^{+/+}$ mice or from intestinal tumors of two APC$^{min/+}$ mice, respectively, were subjected to Western blot analyses using sera (diluted to include 10 μg/ml IgG) from mice having been transferred with PBS alone, Int. B-iGB cells or TiBc-iGB cells as the primary Ab (randomly selected sera from mice in each group in Fig 3B), and HRP-conjugated goat anti-mouse IgG as the secondary Ab (n = 3 per group). The blot was re-probed with Ab against GAPDH, serving as an internal reference protein (lower panels). All data are representative of two independent experiments.

Moutai et al reported that the transferred iGB cells colonized the bone marrow and produced Ab, mainly of the IgG1 class, for several weeks [13]. The results of the experiment shown here revealed that the transfer of iGB cells specific for a surrogate tumor Ag (HEL) suppressed growth of pre-existing (in contrast to simultaneously injected [13]) melanoma cell tumor expressing the same Ag and prolonged the survival of the recipient mice. Tumor inhibition is likely mediated by anti-HEL IgG1 that was produced by the plasma cells derived from

the iGB cells, as Moutai et al reported that such anti-HEL IgG1 bound to HEL-expressing melanoma cells having grown *in vivo*. The mechanism for the antibody-mediated tumor inhibition in this model is currently unclear. Although it is unlikely that the mouse IgG1 Ab mediates the antibody-dependent cellular cytotoxicity (ADCC), but it is possible that it mediates antibody-dependent cellular phagocytosis (ADCP) through binding to FcγRII on phagocytes [27].

TiBcs isolated from intestinal tumors of APC$^{min/+}$ mice, similarly to splenic B cells, could be cultured in iGB cell culture system for a long period by transduction with ER$^{T2}$-Bach2 provided with tamoxifen. Although the TiBc-iGB cells carry a heterozygous *Apc* gene mutation, the extents of their growth rate, class switching and Ab production were similar to those of Int. B-iGB cells derived from APC$^{+/+}$ mice in our culture system. The TiBc-iGB cells were able to produce IgG Abs by withdrawal of tamoxifen and addition of heme, the latter being consistent with published data [12]. However, concentration of Abs produced by the TiBc-iGB cells into the supernatant was variable and often very low on 40LB feeder cells. We solved this problem by using a new feeder cells, 40LA, expressing APRIL in addition to CD40L and BAFF. The iGB cells cultured on the 40LA with heme, IL-5, and IL-6 produced considerable amount of IgG in the supernatant. Thus, we have established a feasible method to expand TiBcs in culture unlimitedly, and to produce considerable amount of Abs from the TiBcs in culture.

The IgG Abs produced from the TiBc-iGB cells could specifically recognize the intestinal tumors in APC$^{min/+}$ mice in immunofluorescence staining assay (Fig 2E). Although we could not evaluate their differentiation into plasma cells or Ab production in mice after transfer, which are indistinguishable from host-derived plasma cells or Abs existing in far excess, inhibition of tumor growth and prolonged survival of APC$^{min/+}$ mice by the transfer of the TiBc-iGB cells are likely mediated by tumor binding of the Abs produced from the transferred TiBc-iGB cells via differentiation into plasma cells. To support this notion, sera from the mice transferred with TiBc-iGB cells contained Abs that selectively recognize some antigens in the tumors of APC$^{min/+}$ mice, as demonstrated by immunofluorescence microscopy. In this experiment (Fig 4A), as well as in the experiment in which supernatant of TiBc-iGB cells was used for staining of tumors of APC$^{min/+}$ mice (Fig 2E), the signals did not appear to be restricted to a particular type of cells in the tumor, although the strength of the signal was heterogenous. This may be plausible if the TiBcs in tumors of individual APC$^{min/+}$ mice are polyclonal and produce Abs against various (neo)antigens in the tumors with various affinities.

Western blot analysis (Fig 4B) demonstrated that the serum Abs in the TiBc-iGB-transferred mice commonly bound to a ~20 kD protein in the extract of tumors of APC$^{min/+}$ mice but not of normal intestines. This data suggests that TiBcs in the tumors formed in individual APC$^{min/+}$ mice recognize a dominant epitope of a protein among many various epitopes, expressed in the APC$^{min/+}$ mouse tumors. As far as we know, no tumor-specific antigens exhibiting epitopes for antibodies in APC$^{min/+}$ mice have been reported, and therefore it has been totally unknown whether individual APC$^{min/+}$ mice or even individual tumors in the same mouse express the same tumor antigens. In this regard, the ~20 kD tumor antigen of APC$^{min/+}$ mice detected by Abs derived from TiBcs in APC$^{min/+}$ mice across the individuals implies the presence of a common tumor neo-antigen recognized by TiBcs among individual APC$^{min/+}$ mice. Such common neo-antigens can be generated by frequent site-specific point mutations, tumor-specific protein modifications, ectopic expression of silent genes, or shared frameshift mutations that have recently been reported to be frequently found in some adenocarcinomas with mismatch repair deficiency and to express immunogenic common neo-peptides in human patients [28]. Identification of such common neo-antigens would lead to generation of therapeutic anti-tumor Abs and anti-tumor vaccines in the model system.

As a next step, we are trying to culture human TiBcs taken from the surgically excised tumor specimens in the iGB cell culture system and to obtain Abs thereof by the similar method as described above. If such Abs bind to the original tumor tissue, we will go on to clonally identify the TiBc-derived iGB cells that produce tumor-specific monoclonal Abs. Such monoclonal Abs would be direct seeds for anti-tumor Ab drugs after clarification of their recognizing (neo-) antigens. Their B cell receptor genes may be applied to generation of chimeric antigen receptor (CAR) for CAR-T cell therapy. From our data, transferring the TiBc-derived iGB cells directly to donor patients appears to be a fascinating alternative of individualized cancer immunotherapies, although further steps of verification should be required for its clinical application.

## Supporting information

**S1 Dataset.**
(XLSX)

**S1 Raw images.**
(PDF)

**S1 File.**
(ZIP)

## Acknowledgments

We thank Dr. Nguyen Tien Dat and Dominika Papiernik (RIBS, Tokyo University of Science) for their contribution to the primary stage of this work, Dr. Tatsuya Moutai (Kaneka corporation) for establishing Bach2-mediated culture system, Dr. Robert Brink (Garvan Institute of Medical Research) for pcDNA3-mHEL plasmid, Drs. Jason G. Cyster (UCSF) and Takaharu Okada (RIKEN IMS) for Hy10 mice, Kei Haniuda, Saori Fukao and other members of Kitamura laboratory for technical advice.

## Author Contributions

**Conceptualization:** Daisuke Kitamura.

**Funding acquisition:** Daisuke Kitamura.

**Investigation:** Xinying Wang.

**Methodology:** Shohei Asami.

**Supervision:** Daisuke Kitamura.

**Writing – original draft:** Xinying Wang.

**Writing – review & editing:** Daisuke Kitamura.

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
