## [Decision Letter · Decision Letter 0]

9 Jul 2020

PONE-D-20-19053

A novel cancer immunotherapy using tumor-infiltrating B cells in the APC (min/+) mouse model

PLOS ONE

Dear Dr. Kitamura

Thank you for submitting your manuscript to PLOS ONE. After careful consideration, we feel that it has merit but does not fully meet PLOS ONE’s publication criteria as it currently stands. Therefore, we invite you to submit a revised version of the manuscript that addresses the points raised during the review process.

While both reviewers feel that the study is interesting, they have several concerns about the authors’ results and experimental conditions. Specifically, histological analysis is very poor and not sufficient to support the authors’ conclusion. Moreover, it is unclear why the authors only focus on IgG, but not IgM or IgA. Finally, the mechanism how TiBcs reduce the size and numbers of tumor is not investigated.

We look forward to receiving your revised manuscript.

Kind regards,

Hiroyasu Nakano, M.D., Ph.D.

Academic Editor

PLOS ONE

Journal Requirements:

Additional Editor Comments (if provided):

Reviewers' comments:

Reviewer's Responses to Questions

**Comments to the Author**

1. Is the manuscript technically sound, and do the data support the conclusions?

Reviewer #1: Yes

Reviewer #2: Partly

2. Has the statistical analysis been performed appropriately and rigorously? 

Reviewer #1: Yes

Reviewer #2: I Don't Know

3. Have the authors made all data underlying the findings in their manuscript fully available?

Reviewer #1: Yes

Reviewer #2: Yes

4. Is the manuscript presented in an intelligible fashion and written in standard English?

Reviewer #1: Yes

Reviewer #2: Yes

5. Review Comments to the Author

Reviewer #1: The authors clearly demonstrated that TIBCs, which have been expanded on 40LB system, suppressed intestinal tumor growth and elongated the survival of recipient APCmin/+ mice.

There are several basic questions or concerns, which would be better if they could be answered or discussed in the manuscript.

1. As compared to the infusion experiment of HEL-specific iGB cells into mice, what is the effect of HEL-specific iGB cells on tumor cells in in vitro culture? Please describe the mechanism how TIBCs attack tumor cells in vitro and in vitro. Which is important to suppress the tumor growth, B cells or the antibodies produced by TIBCs? The authors mentioned ‘by a similar mechanism as monoclonal Ab drugs in vivo [27]’, in line 413. ‘Similar’ is not enough scientific.

2. In Figure 2, the authors showed that significant number of IgM/IgA positive cells could be expanded on their 40LB culture system. However, they ignored IgM/IgA antibodies. Please discuss why. Also, in Figure 1A, the number of population size (%) should be moved out of dots. In the same figure, unnecessary small characters (such as <pe-cy7-a> IgM, etc.) should be cleared.

3. In Figures 2D and 4B, FITC signals were too damn to identify which cells were stained, tumor cell or epithelial cells. Please replace the data with clearer ones or magnified ones. Otherwise, I do not think convincing.

4. In Figures 2D, 4B and 4C, please specify the origins of antibodies and tissue sections. Were they always derived from the same individual? Is it already widely accepted that any APCmin/+ mice express the same tumor antigen and produce the antibodies against the same antigen? Individual difference should be considered or discussed.

5. Figure 4C, please explain why IgM/IgA antibodies were ignored.

6. In line 183, ‘mg/ml‘ should be ‘�g/ml’. In line 195, ‘6-mm-thick’ should be ‘6-�m-thick’.

7. This is a naïve question. Did TIBC-iGB cells successfully infiltrate into tumor? It would be better to have such data to convince the general readers.</pe-cy7-a>

Reviewer #2: Functional roles of tumor-infiltrating T cells in tumor expansion or regression have been extensively investigated. However, their roles of tumor-infiltrating B cells are limited. In the previous study, the authors’ group has developed a system to efficiently expand primary B cells in vitro. Using this system, in this manuscript, Wang et al. investigate the effect of tumor-infiltrating B cells on tumor growth in vivo. Repeated injection of tumor-infiltrating B cells into Apcmin/+ mice dramatically prolongs their survival and reduces the size and numbers of tumors. Moreover, the authors show that numbers of IgG-positive cells appear in the tumor tissues of Apcmin/+ mice. Although the study presented here is interesting and deserves for publication, there are several concerns need to be addressed before publication. The followings are specific comments.

Major pints:

1. TIBCs does not appear to be appropriate for abbreviation of tumor-infiltrating B cells, since TIBCs are frequently used for abbreviation of total iron binding capacities. The reviewer recommends that the authors might use TiBcs instead of TIBCs (Linnebacher et al, Oncoimmunology, 1:1186-1188).

2. In Figures 2D, 3E, and 4B, it is unclear whether B cells accumulate around the tumor tissues. The authors need to show more representative images of hematoxylin-eosin staining of normal and tumor tissues of the small intestines. Moreover, the authors need to perform immunohistochemistry using antibody against a specific marker of B cells (B220).

3. In Figures 2 to 4, the authors isolate and expand tumor-infiltrating B cells from Apcmin/+ mice. Since B cells also harbor Apc mutation, the authors need to discuss the possibility that the mutation of Apc gene might affect the function of TiBcs.

4. In Figures 2D and 4B, the immunofluorescent analysis of the tissues stained with the culture supernatants or murine sera is not conclusive. It is hard to speculate which cells, such as tumor cells, infiltrated B cells, or normal epithelial cells are stained with the antibodies. Thus, the authors need to show more representative images. Ideally, the authors might perform double immunostaining with the culture supernatant or murine sera along with antibodies against a specific marker of B cells (B220) and epithelial cells (E-cadherin).

5. In Figure 3, the effect of injection of TiBcs on survival of Apcmin/+ mice appears to be dramatic. However, numbers of mice analyzed in Figure 3D are not sufficient to draw conclusion. To further verify the authors’ conclusion in Figure 3D, the authors need to increase numbers of mice (more than 4 or 5 mice per each group). It is unclear how injection of TiBcs reduces the size of tumors. To address this issue, the authors need to quantify numbers of tumor cells undergoing apoptosis (active caspase 3-positive cells) and numbers of proliferating tumor cells (Ki67-positive cells).

6. In the Material and Methods and Figure legends, they have to describe how they performed the statistical analysis.

Minor points:

1. In Figure 2A, the size of characters representing percentages of cell populations is very small. The authors need to change them to more larger ones.

2. The authors need to include scale bars in all images of histological analysis.

3. In Materials and Methods, clone names or catalog numbers of the antibodies used in the study should be mentioned.

4. In Figure 4C, the authors need to include the results of Western blotting with anti-tubulin or beta-actin to ensure equal loading of each sample.

6. PLOS authors have the option to publish the peer review history of their article (what does this mean?). If published, this will include your full peer review and any attached files.

Reviewer #1: **Yes: **Reiko Shinkura

Reviewer #2: No

---

## [Author Response · Author response to Decision Letter 0]

31 Dec 2020

Reviewer #1: 

The authors clearly demonstrated that TIBCs, which have been expanded on 40LB system, suppressed intestinal tumor growth and elongated the survival of recipient APCmin/+ mice.

There are several basic questions or concerns, which would be better if they could be answered or discussed in the manuscript.

Comment 1: As compared to the infusion experiment of HEL-specific iGB cells into mice, what is the effect of HEL-specific iGB cells on tumor cells in in vitro culture? Please describe the mechanism how TIBCs attack tumor cells in vitro and in vitro. Which is important to suppress the tumor growth, B cells or the antibodies produced by TIBCs? The authors mentioned ‘by a similar mechanism as monoclonal Ab drugs in vivo [27]’, in line 413. ‘Similar’ is not enough scientific.

Our response: In our previous report (PLOS ONE 2014, 9:e92732; [13]), we demonstrated that the same HEL-specific iGB cells develop into bone-marrow plasma cells and produce HEL-specific IgG1 (and IgG2b in low abundance) in vivo after being injected into mice. In addition, when B16-mHEL cells were also injected i.v., the anti-HEL IgG1 bound to the surface of B16-mHEL cells in excised lung tumors and suppressed tumor growth in the lungs. Therefore, we suppose that antibody should play a main role for this tumor suppression, also in the current experiment shown in the Fig 1. In both experiments, the HEL-specific iGB cells, generated from spleen B cells initially stimulated with IL-4, dominantly produced IgG1 in vivo. IgG1-mediated tumor suppression is not likely mediated by ADCC or CDC but might be by the antibody-dependent cellular phagocytosis (ADCP), as we have described in the Discussion section of the revised manuscript (page 24, line 482~485).

The suggested in-vitro experiment would be that B16-mHEL cells are incubated with HEL-specific IgG1 and effecter cells such as macrophages or neutrophils. Even if we would find a condition in which the B16-mHEL cells are killed by ADCP, it would not prove that is happening in vivo. Elucidation of the mechanism for the iGB cell-mediated tumor suppression in vivo would need various kinds of experiments that might not always show clear evidence, which we think would be a subject on the next stage of this project. 

Comment 2: In Figure 2, the authors showed that significant number of IgM/IgA positive cells could be expanded on their 40LB culture system. However, they ignored IgM/IgA antibodies. Please discuss why. Also, in Figure 1A, the number of population size (%) should be moved out of dots. In the same figure, unnecessary small characters (such as IgM, etc.) should be cleared. 

Our response: In order to obtain more abundant Abs from the iGB cells, we have changed the culture method for the Ab production, namely using 40LA feeder cells with IL-5 and IL-6 (described in the revised Materials and Methods section). This allowed production of relatively large amount of IgG (6~7 μg/ml) but little IgA or IgM (30~40 ng/ml) in the supernatant (the new Fig 2D), as described in the revised manuscript (page 18, line 347~349), although the reason for the inefficient IgA/IgM production is unclear. Therefore, we have used only anti-mouse IgG as the secondary Ab for the sections primarily stained with the supernatants (the new Fig 2E).

 We have revised the former Fig 2A (supposedly meant by the reviewer) according to the reviewer’s comment, as shown in the new Fig 2B.

Comment 3: In Figures 2D and 4B, FITC signals were too damn [dim?] to identify which cells were stained, tumor cell or epithelial cells. Please replace the data with clearer ones or magnified ones. Otherwise, I do not think convincing. 

Our response: To improve the staining data in the former Fig 2D, we have used new iGB cell culture supernatants containing a higher concentration of IgG, as stated above. As a result, the FITC signal became brighter and specific to the tumor region, except a few spots in the normal intestinal tissue and other control staining, which may represent IgG+ plasma cells detected by the secondary anti-mouse IgG Ab. 

 As for the former Fig 4B, we stained the sections with sera of iGB-cell-transferred mice containing 10 μg/ml of IgG, in which the concentration of the iGB-cell-derived IgG was unknown (indistinguishable from endogenous IgG of the host) and supposedly low. Therefore, we have used undiluted sera in the new experiment, which has improved the staining to some extent as shown in the new Fig 4A.

 In both experiments, the signals did not appear to be restricted to a particular type of cells in the tumor, although the strength of the signal was heterogenous. This may be plausible if the TiBcs in tumors of individual APCmin/+ mice are polyclonal and produce Abs against various (neo)antigens in the tumors with various affinities. This notion has now been described in the Discussion section of the revised manuscript (page 25, line 511~517).

Comment 4: In Figures 2D, 4B and 4C, please specify the origins of antibodies and tissue sections. Were they always derived from the same individual? Is it already widely accepted that any APCmin/+ mice express the same tumor antigen and produce the antibodies against the same antigen? Individual difference should be considered or discussed. 

Our response: In the former Figs 2D, 4B, 4C, the origins of iGB cells producing Abs and those of tissue sections were different. In the corresponding new Figs 2E, 4A, 4B, as well as in the new Figs 3, Abs were derived form TiBc-iGB cells originated from mixed TiBcs of three APCmin/+ mice, or from Int. B-iGB cells originated from mixed intestinal B cells of three APC+/+ mice. The tissue sections and extracts for Western blotting were from individuals that were different from the origins of the Abs.

 As far as we know, no tumor antigens recognized by any antibodies produced in APCmin/+ mice have been reported, and we are the first to derive Abs from TiBc in the tumor of APCmin/+ mice. Therefore, it is totally unknown whether individual APCmin/+ mice or even individual tumors in the same mouse express the same tumor antigens for antibodies. In this regard, the ~20 kD protein expressed in the tumors of APCmin/+ mice detected by the sera of mice transferred with TiBc-iGB cells (originated from different individuals) implies that the presence of a tumor antigen commonly recognized by TiBcs of individual APCmin/+ mice. Such common neo-antigens can be generated by frequent site-specific point mutations, tumor-specific protein modifications, ectopic expression of silent genes, or shared frameshift mutations that has recently been reported to be frequently found in some adenocarcinomas with mismatch repair deficiency and express immunogenic common neo-peptides in human patients (Cell 2020, 183:1634; [28]). This consideration has now been included in the Discussion section of the revised manuscript (page 25, line 520~536).

Comment 5: Figure 4C, please explain why IgM/IgA antibodies were ignored. 

Our response: We have performed Western blotting using the same protein samples and the same set of primary Abs (sera) as in the new Fig 4B (corresponding to the former Fig 4C) with anti-IgA or anti-IgM Abs as the secondary Abs. As a result, no tumor-specific bands were selectively detected with the sera of mice injected with TiBc-iGB cells. Therefore, we have not included these data in the revised manuscript but deposited in the Supporting Information for reference.

Comment 6: In line 183, ‘mg/ml‘ should be ‘μg/ml’. In line 195, ‘6-mm-thick’ should be ‘6-μm-thick’. 

Our response: We have corrected these errors.

Comment 7: This is a naïve question. Did TIBC-iGB cells successfully infiltrate into tumor? It would be better to have such data to convince the general readers. 

Our response: We think that the transferred TiBc-iGB cells need not infiltrate into tumors to exert anti-tumor activity, as they differentiate into plasma cells somewhere such as the bone marrow and produce anti-tumor Abs, as we previously reported for iGB cells derived from spleen B cells ([13]; mentioned in page 23, line 474~475 in the revised manuscript). Therefore, we do not think that the suggested data is necessary.

Reviewer #2: 

Functional roles of tumor-infiltrating T cells in tumor expansion or regression have been extensively investigated. However, their roles of tumor-infiltrating B cells are limited. In the previous study, the authors’ group has developed a system to efficiently expand primary B cells in vitro. Using this system, in this manuscript, Wang et al. investigate the effect of tumor-infiltrating B cells on tumor growth in vivo. Repeated injection of tumor-infiltrating B cells into Apcmin/+ mice dramatically prolongs their survival and reduces the size and numbers of tumors. Moreover, the authors show that numbers of IgG-positive cells appear in the tumor tissues of Apcmin/+ mice. Although the study presented here is interesting and deserves for publication, there are several concerns need to be addressed before publication. The followings are specific comments.

Major pints:

Comment 1: TIBCs does not appear to be appropriate for abbreviation of tumor-infiltrating B cells, since TIBCs are frequently used for abbreviation of total iron binding capacities. The reviewer recommends that the authors might use TiBcs instead of TIBCs (Linnebacher et al, Oncoimmunology, 1:1186-1188).

Our response: We have replaced all the ‘TIBCs’ with ‘TiBcs’ in the revised manuscript.

Comment 2: In Figures 2D, 3E, and 4B, it is unclear whether B cells accumulate around the tumor tissues. The authors need to show more representative images of hematoxylin-eosin staining of normal and tumor tissues of the small intestines. Moreover, the authors need to perform immunohistochemistry using antibody against a specific marker of B cells (B220).

Our response: We have now replaced the former Figs 2D, 3E and 4B with the corresponding new Figs 2E, 3E, and 4A. Since it was difficult to identify B cells in the hematoxylin-eosin (HE) staining images, we have performed immunofluorescent staining with anti-B220 antibody of intestinal tissue including normal and tumor areas of APCmin/+ mice. As shown in the new Figs 2A, small clusters of B cells sparsely scattered mainly in the interstitial regions of the tumor and in the lamina propria of normal mucosa. We have stated this in the revised manuscript (page 16, line 304-306). 

Comment 3: In Figures 2 to 4, the authors isolate and expand tumor-infiltrating B cells from Apcmin/+ mice. Since B cells also harbor Apc mutation, the authors need to discuss the possibility that the mutation of Apc gene might affect the function of TiBcs.

Our response: Although the heterozygous Apc gene mutation in B cells might affect some aspect of their function, we observed similar extents of proliferation, class switching, and Ab production between TiBc-iGB cells derived from APCmin/+ mice and Int.B-iGB cells from APC+/+ mice in our culture system (described in the Discussion section of the revised manuscript; page 24, line 488~491). We could not evaluate the in vivo growth of the transferred iGB cells or their differentiation into plasma cells since they are indistinguishable from the host B cells or plasma cells existing in far excess. In addition, we could not quantify the Abs produced from the iGB cells since they are indistinguishable from the Abs produced by the host mice (partly mentioned in the Discussion section; page 25, line 504~506). Therefore, it is almost impossible to assess the in vivo effect of the APC mutation on the iGB cells after transfer into mice. At least, lymphoma did not develop in the mice transferred with TiBc-iGB cells, as supported by their better survival than the mice transferred with Int. B-iGB cells. 

Comment 4: In Figures 2D and 4B, the immunofluorescent analysis of the tissues stained with the culture supernatants or murine sera is not conclusive. It is hard to speculate which cells, such as tumor cells, infiltrated B cells, or normal epithelial cells are stained with the antibodies. Thus, the authors need to show more representative images. Ideally, the authors might perform double immunostaining with the culture supernatant or murine sera along with antibodies against a specific marker of B cells (B220) and epithelial cells (E-cadherin). 

Our response: We have performed again the immunofluorescent staining as shown in the new Figs 2E and 4A (corresponding the former 2D and 4B). Serial sections of intestines of APCmin/+ mice including normal and tumor area have been stained with the supernatants (Fig 2E) or the mouse sera (Fig 4A), or anti-E-Cadherin, together with DAPI. We think the resulting data have been significantly improved. Since the anti-E-Cadherin Ab that stained well among those we tested was of mouse IgG2a/κ (BD transduction laboratory 610182; cross-reacting to mouse E-Cadherin) that needs secondary anti-mouse IgG Ab, we could not do the multiple staining as the reviewer suggested. To show the presence of B cells in the tumor of APCmin/+ mice, we have performed double staining for B220 and E-Cadherin with DAPI as shown in the new Fig 2A (though the E-Cadherin staining was weak in this combination). 

 As stated in the response to the Comment 3 of the Reviewer #1, the signals did not appear to be restricted to a particular type of cells in the tumor, although the strength of the signal was heterogenous, in both experiments. This may be plausible if the TiBcs in tumors of individual APCmin/+ mice are polyclonal and produce Abs against various (neo)antigens in the tumors with various affinities. This notion has now been described in the Discussion section of the revised manuscript (page 25, line 511~517).

Comment 5: In Figure 3, the effect of injection of TiBcs on survival of Apcmin/+ mice appears to be dramatic. However, numbers of mice analyzed in Figure 3D are not sufficient to draw conclusion. To further verify the authors’ conclusion in Figure 3D, the authors need to increase numbers of mice (more than 4 or 5 mice per each group). It is unclear how injection of TiBcs reduces the size of tumors. To address this issue, the authors need to quantify numbers of tumor cells undergoing apoptosis (active caspase 3-positive cells) and numbers of proliferating tumor cells (Ki67-positive cells).

Our response: According to the reviewer’s requested, we have newly performed the experiment with increased number of mice (n=5 for the new Fig 3D), setting control mice transferred with iGB cells derived from normal intestinal B cells of APC+/+ mice (Int. B-iGB) or PBS alone (the same controls as in Fig 3B, C). The result clearly showed that injection of TiBc-iGB cells reduce the size of tumors in APCmin/+ mice (the new Fig 3D).

 To address the mechanism, we have performed immunohistochemistry analysis of the tumors with Abs against active (cleaved) caspase 3 and Ki67. We could not quantify the number of positive cells because of their various and uneven distribution throughout the intestines, and we also think that the positive number would not always corelate with the effect on tumor cells because they do not include already dead and disappeared cells. Nevertheless, the result has demonstrated that active caspase 3 staining appears stronger and Ki67 staining weaker in the tumors of mice transferred with TiBc-iGB cells (the new Fig 3E), suggesting that the tumor cells were dying through apoptosis and their growth was inhibited by Abs produced from the transferred TiBc-iGB cells. 

 We have described this result in the Result section (page 20, line 394~397) and the method for the immunohistochemistry in the Materials and Methods section (page 12, line 227~239) in the revised manuscript.

Comment 6: In the Material and Methods and Figure legends, they have to describe how they performed the statistical analysis. 

Our response: We have described about the method of statistical analysis in the Material and Methods section (page 13, line 250~254) and the relevant parts of the Figure legends of the revised manuscript.

Minor points:

1. In Figure 2A, the size of characters representing percentages of cell populations is very small. The authors need to change them to more larger ones.

Our response: We have enlarged the font size of the numbers and put them out of the windows in the new Fig 2B (corresponding to the former Fig 2A).

2. The authors need to include scale bars in all images of histological analysis.

Our response: We have included the scale bars in all the images in the new Figs 2A, 2E, 3E, and 4A.

3. In Materials and Methods, clone names or catalog numbers of the antibodies used in the study should be mentioned. 

Our response: The clone names for the Abs have been included in Materials and Methods section of the revised manuscript.

4. In Figure 4C, the authors need to include the results of Western blotting with anti-tubulin or beta-actin to ensure equal loading of each sample. 

Our response: For the sample loading control, the Western blot has been re-probed with anti-GAPDH antibody that we have been using for that purpose.

---

## [Decision Letter · Decision Letter 1]

5 Jan 2021

A novel cancer immunotherapy using tumor-infiltrating B cells in the APCmin/+ mouse model

PONE-D-20-19053R1

Dear Dr. Kitamura

We’re pleased to inform you that your manuscript has been judged scientifically suitable for publication and will be formally accepted for publication once it meets all outstanding technical requirements.

Kind regards,

Hiroyasu Nakano, M.D., Ph.D.

Academic Editor

PLOS ONE

Additional Editor Comments (optional):

Reviewers' comments:

Reviewer's Responses to Questions

**Comments to the Author**

1. If the authors have adequately addressed your comments raised in a previous round of review and you feel that this manuscript is now acceptable for publication, you may indicate that here to bypass the “Comments to the Author” section, enter your conflict of interest statement in the “Confidential to Editor” section, and submit your "Accept" recommendation.

Reviewer #1: All comments have been addressed

Reviewer #2: All comments have been addressed

2. Is the manuscript technically sound, and do the data support the conclusions?

Reviewer #1: Yes

Reviewer #2: Yes

3. Has the statistical analysis been performed appropriately and rigorously? 

Reviewer #1: Yes

Reviewer #2: Yes

4. Have the authors made all data underlying the findings in their manuscript fully available?

Reviewer #1: Yes

Reviewer #2: Yes

5. Is the manuscript presented in an intelligible fashion and written in standard English?

Reviewer #1: Yes

Reviewer #2: Yes

6. Review Comments to the Author

Reviewer #1: In the revised version, the authors discussed all the raised points logically and well-discussed on limitations of experiments. Now it is ready for publication.

Reviewer #2: (No Response)

7. PLOS authors have the option to publish the peer review history of their article (what does this mean?). If published, this will include your full peer review and any attached files.

Reviewer #1: No

Reviewer #2: No

---

## [Editor Report · Acceptance letter]

7 Jan 2021

PONE-D-20-19053R1 

A novel cancer immunotherapy using tumor-infiltrating B cells in the APC^min/+^ mouse model 

Dear Dr. Kitamura:

I'm pleased to inform you that your manuscript has been deemed suitable for publication in PLOS ONE. Congratulations! Your manuscript is now with our production department. 

Kind regards, 

on behalf of

Professor Hiroyasu Nakano 

Academic Editor

PLOS ONE